# Curve Skeleton Extraction from Incomplete Point Clouds of Livestock and Its Application in Posture Evaluation

Yihu Hu [1], Xinying Luo [1], Zicheng Gao [1], Ao Du [1,2], Hao Guo [1,2,*], Alexey Ruchay [3,4], Francesco Marinello [5] and Andrea Pezzuolo [5]

1. College of Land Science and Technology, China Agricultural University, Beijing 100083, China; huyihu@cau.edu.cn (Y.H.); luoxy@cau.edu.cn (X.L.); gaozicheng@cau.edu.cn (Z.G.); s20193081375@cau.edu.cn (A.D.)
2. College of Information and Electrical Engineering, China Agricultural University, Beijing 100083, China
3. Federal Research Centre of Biological Systems and Agro-Technologies of the Russian Academy of Sciences, 460000 Orenburg, Russia; ran@csu.ru
4. Department of Mathematics, Chelyabinsk State University, 454001 Chelyabinsk, Russia
5. Department of Land, Environment, Agriculture and Forestry, University of Padova, Viale dell'Universit´a 16, 35020 Legnaro, Italy; francesco.marinello@unipd.it (F.M.); andrea.pezzuolo@unipd.it (A.P.)
* Correspondence: guohaolys@cau.edu.cn

**Abstract:** As consumer-grade depth sensors provide an efficient and low-cost way to obtain point cloud data, an increasing number of applications regarding the acquisition and processing of livestock point clouds have been proposed. Curve skeletons are abstract representations of 3D data, and they have great potential for the analysis and understanding of livestock point clouds. Articulated skeleton extraction has been extensively studied on 2D and 3D data. Nevertheless, robust and accurate skeleton extraction from point set sequences captured by consumer-grade depth cameras remains challenging since such data are often corrupted by substantial noise and outliers. Additionally, few approaches have been proposed to overcome this problem. In this paper, we present a novel curve skeleton extraction method for point clouds of four-legged animals. First, the 2D top view of the livestock was constructed using the concave hull algorithm. The livestock data were divided into the left and right sides along the bilateral symmetry plane of the livestock. Then, the corresponding 2D side views were constructed. Second, discrete skeleton evolution (DSE) was utilized to extract the skeletons from those 2D views. Finally, we divided the extracted skeletons into torso branches and leg branches. We translated each leg skeleton point to the border of the nearest banded point cluster and then moved it to the approximate centre of the leg. The torso skeleton points were calculated according to their positions on the side view and top view. Extensive experiments show that quality curve skeletons can be extracted from many livestock species. Additionally, we compared our method with representative skeleton extraction approaches, and the results show that our method performs better in avoiding topological errors caused by the shape characteristics of livestock. Furthermore, we demonstrated the effectiveness of our extracted skeleton in detecting frames containing pigs with correct postures from the point cloud stream.

**Keywords:** curve skeleton; posture; pig; point cloud; precision livestock farming

## 1. Introduction

With consumer-grade depth sensors currently applied in farm environments [1,2], point clouds, a type of 3D data structure, show many advantages for body measurement that traditional 2D pixel data do not possess [3,4]. An increasing number of studies on the scanning and processing of livestock point cloud data have been developed [5–10]. The 3D scanning systems for acquiring the point cloud streams of livestock body surfaces from complex farm environments have been implemented in different studies [11–14]. Livestock body measurement and lameness detection [15,16] can be further investigated

based on these kinds of systems. However, for tasks that utilize only some of the properties of the original shape of the animal, storing and processing whole livestock body surface datasets is unnecessary. Using traits to estimate body measurements has appeared in many studies [17–21].

Skeletons, as abstract representations that jointly describe the geometry, topology, and symmetry properties of a shape in compact and intuitive ways, are beneficial in modelling and animation applications [22]. Choosing frames that contain suitable postures for animal body measurement can be implemented efficiently with the assistance of the skeleton. Numerous skeleton extraction algorithms have been proposed. Numerous methods of extracting skeletons from 2D shapes have been extensively investigated, and these methods lead to satisfactory results. Bai et al. proposed robust methods [23,24] for pruning redundant branches, and these are used as part of our algorithm. Skeleton extraction algorithms for 3D shapes are more time consuming than 2D approaches due to the highly complex spatial relationships in 3D data. Skeletons of 3D shapes can be roughly divided into surface skeletons and curve skeletons. Curve skeletons of tubular shapes with local, axial, circular cross-sections can be defined as 1D structures that, at least, preserve the original shape topology [22].

Tagliasacchi [25] proposed a mean curvature skeleton for 3D mesh data. However, approaches such as this [25,26], which are based on mesh contractions, are applied to watertight data. In farm environments, it is difficult to obtain such high-quality point clouds due to occlusion on the inside of the animal body and missing data[5,7]. The most feasible methods are those of [27,28] because these can directly extract a skeleton from an incomplete point cloud. They calculate skeleton points from local point sets as branches and then connect various branches to form the curve skeleton. Skeletons extracted from these methods can be applied in various applications, such as shape matching, deformation, and retrieval [29,30]. The calculation of the rotational symmetry axis proposed in [27] relies on a cylindrical shape prior and accurate point normals. The $L_1$-median skeleton approach [28] is a representative skeletonization method that can directly project point samples onto their local centres without building a point connectivity or estimating point normals. However, different branches of livestock point clouds calculated by the $L_1$-median approach need to be extracted using different parameters, making the process unstable and time consuming. The skeleton transfer method proposed by Seylan et al. [31] avoids the time losses incurred by extracting all the skeletons from a sequence and finds the correspondence between the curve skeletons of each frame. However, the skeleton source that is used for the transfer still needs to be extracted by a skeleton extraction method. Point2skeleton [32] is a unsupervised skeleton extraction method which can learn skeletal representations from point clouds. These two methods [28,32] can be implemented on point clouds of generic shapes (including livestock) and will therefore be compared with our method in this paper.

Common skeleton extraction methods are designed for a general shape, lacking the priors of topology. Therefore, a top-down design is most suitable for livestock point clouds. Knowing that most livestock have fixed topologies and symmetric structures, the speed and robustness of the extraction approaches can be improved. There are already some studies [33–35] utilizing prior knowledge about human structures to estimate joint positions or calculate the skeletons of the original shapes. Barros [34] define a skeleton model and fit the skeleton to its proper corresponding body parts. Li [35] added length conservation and symmetry constraints to improve the accuracy of the extracted skeleton. The skeleton extraction approach for plants has also been proposed in the field of plant phenotyping [36,37]. However, there are no studies focusing on a skeleton extraction method specifically for livestock.

In this paper, we propose a novel method that utilizes an existing 2D skeleton extraction method and the priors of livestock to extract the curve skeletons of livestock. Since the skeleton is calculated according to the contour of a projected point cloud, the time-consuming process of calculating a large number of point clouds is avoided. According

to [38], without motion-related cues, the skeleton extracted by general methods [27,28,39] is the abstract shape of a single static model and cannot be applied to other applications. The proposed method relies on pose normalization [40], and the extracted curve skeleton can be mapped to the same coordinate system. Additionally, our extracted skeleton is divided into multiple branches, including four leg branches and a torso branch. We sort the skeleton points on each branch by their spatial relationships, and the target skeleton points of a specific branch are retrieved according to its approximate position. The resulting skeleton has excellent potential for the detection of body measurement landmarks and motion-related tasks, such as lameness detection. The main contributions of this paper are as follows:

The prior knowledge of animal body is introduced into animal skeleton estimation with an original spatial-relationship branch classification. A novel curve skeleton extraction method for four-legged livestock is proposed. In comparison with the skeleton extraction approaches [28,32], the proposed algorithm performs robustly and efficiently on incomplete livestock point cloud data. Based on the extracted skeleton, a method of evaluating the correct posture [6] for given body measurements is proposed.

## 2. Materials and Methods

### 2.1. Experimental Data

The experimental data consist of datasets scanned in farm environments and a synthetic dataset of other species. The first dataset contains point clouds of live pigs in the real world, scanned from the ShangDong WeiHai swine-breeding centre of DA BEI NONG GROUP, with ages ranging from 130 to 220 days. These pigs have long bodies and short hair, with heights between 52 and 66 centimetres(cm) and lengths varying from 80.5 to 104 centimetres(cm). Twenty Landrace pig sequences were randomly chosen to make up the real-world point cloud dataset, and each sequence contained 10 pig point clouds with different postures. We used the prototype scanning system [6], which consists of two calibrated Xtion Pro Live sensors. The two sensors from two different viewpoints can cover the entire body surface of livestock. Live 3D point cloud sequences were captured by these two sensors simultaneously. The frame rate is about 4 FPS(Frames Per Second). Figure 1 shows a raw scanned pig. A total of 200 point clouds compose our first dataset. These pig data were scanned in a pig house, and the obtained point clouds contain pigs, floors, and farm facilities. The pig being scanned was protected from direct sunlight during the data collection process. Although the point cloud quality changed slightly with distance, the effect of the distance from the Xtion Pro sensor to the pigs was not investigated. The scanning distance ranges from 0.8 to 1.6 m. The average point spacing of these point clouds $r$ is set to 0.005 m.

The synthetic dataset containing animal point clouds of other species, including 3 pigs, 2 horses, 3 cows, 2 hippos, 2 rhinos, and 2 water buffalos. Our data has many features. First of all, the data are collected by Xtion Pro. This consumer grade depth sensor ensures the low cost of data acquisition, but there are some noises in the data set, which can test the robustness of our method. Secondly, data acquisition is not affected by lighting conditions. Finally, the data contains pig bodies with different postures photographed from different angles and distances. This variability can enable us to design methods with stronger generalization ability. We use these data to verify the applicability of our method to different species of four-legged animals. There are differences in size between these datasets, and we normalize the data size by scaling its bounding box. Our experiments are implemented on the data after pose normalization.

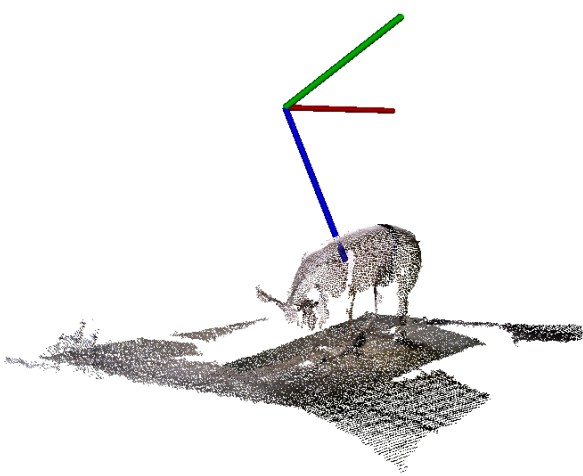

**Figure 1.** The visualization of a raw scanned pig.

### 2.2. Livestock Data Pre-Processing

To design the algorithm and indicate the applicable data range, some assumptions about the input 3D animal data need to be stated in advance. We apply the assumptions about data acquisition and normalization proposed in [5,40]. We assume that the posture of the animal being scanned is "standing". The shape of an animal is mainly composed of its head, body, and limbs, indicating that the topological structure of applicable animal data is fixed. Additionally, to make different body parts of the livestock distinguishable, some special postures should be avoided as much as possible. For example, the animal completely bends its head against its legs or the legs on different sides of the body are crossed. More assumptions are detailed in [5,40].

A rigorous description of the technical background point cloud processing is available in [41]. Here, we briefly outline the pre-processing pipeline, which enables this work to be reproducible. First, we needed to extract livestock from the background of the point cloud. We detected the ground plane by the random sample consensus (RANSAC) algorithm [42] and removed it. Once we performed this operation, the livestock were spatially separated from other data in the point cloud of the scene. We then used region growing to obtain a set of clusters. As the raw scanned data mainly consisted of one livestock standing on a planar ground with possible components of other livestock facilities, we could infer that the livestock data were the largest cluster. After we extracted the livestock from the raw scanned data, each livestock was in an arbitrary orientation and position in three-dimensional space. Then, we applied the pose normalization algorithm to align the segmented livestock with the canonical coordinate system (CCS) [40].

- The origin of the CCS is set as the centroid of the livestock.
- The Z-axis is perpendicular to the bilateral symmetry plane, and its positive side points to the right side of the body.
- The Y-axis is perpendicular to the ground plane, and its positive direction points to the dorsal of the livestock.
- The X-axis is perpendicular to the plane that consists of the Z-axis and Y-axis, and its positive direction goes from the origin of the CCS to the head of the livestock.

The bilateral symmetry plane of livestock was estimated by using a voting scheme. Specifically, we pruned a point set from the body part roughly located on the same horizontal plane. A pair of points in the pruned point set stand for a vote. We chose a set of one-to-many samples from the pruned point clouds, applied the global symmetric transformations induced by these samples, and estimated the bilateral symmetry plane based on the majority of votes. The *y*-axis was determined by the normal vector of the detected ground plane. The forward direction of the *x*-axis was identified using the application-specific

geometric features of livestock. After preparing all the components above, we estimated a 3D rigid alignment for performing pose normalization on the livestock (for more details about segmentation and pose normalization, please refer to [6,40]). Figure 2 shows a pig registered in the CCS.

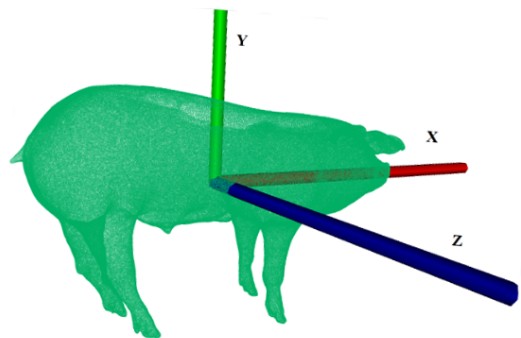

**Figure 2.** The visualization of a pig registered in the canonical coordinate system (CCS).

*2.3. Curve Skeleton Extraction*

Bilateral symmetry planes are almost universal among animals, and they provide a special perspective for analysing the topological structures of livestock. On a four-legged livestock with fixed topology, the legs are located on both sides of the symmetry plane, and the torso skeleton points are distributed near the symmetry plane. For each half of the livestock data (divided by the symmetry plane), we observed that the contour of its projection on the symmetry plane effectively represents the shape of livestock. The processing pipeline for skeleton extraction is shown in Figure 3. The first step of our curve skeleton extraction method was the definition of the side 2D views for each half of the livestock. Then, the side views of the skeleton were extracted using the discrete skeleton evolution (DSE) algorithm. The extracted skeleton was divided into leg branches and torso branches. For the torso skeleton point, we combined the side views and top view of the torso skeleton to calculate its final position. For the leg branch, we moved the leg skeleton to the border of the livestock as a reasonable initial position. Then, we constructed the nearest banded shape cluster for each leg skeleton point and moved each leg skeleton point to the target position according to its corresponding cluster. Essentially, since the leg skeleton points are only translated along the z-axis in the subsequent process, the centrality of the leg skeleton is well preserved.

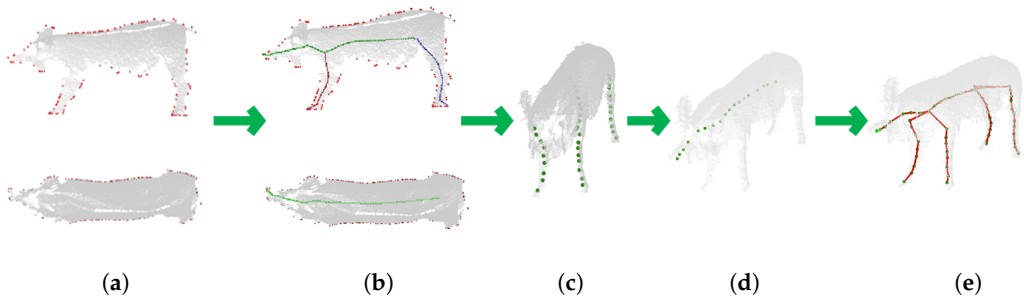

(**a**)          (**b**)          (**c**)          (**d**)          (**e**)

**Figure 3.** A visual diagram of the processing pipeline for curve skeleton extraction: (**a**) Contour construction, (**b**) Skeleton extraction and division, (**c**) Leg skeleton calculation, (**d**) Torso skeleton calculation and (**e**) Final skeleton.

2.3.1. Construct the Contours of the Side Views

We started with the normalized livestock point cloud denoted by $S = \{p_i\}$. The bilateral symmetry plane denoted by $P_s$ divided the livestock into two parts: the left side

of the body denoted by $S_l$ and the right side of the body denoted by $S_r$. $S_l = \{p_i | p_i.z < 0, p_i \in S\}$ and $S_r = \{p_i | p_i.z > 0, p_i \in S\}$ were estimated after filtering. Moreover, both sides of the body were treated separately. The process was the same for $S_l$ and $S_r$. First, we projected $S_l$ to the symmetry plane $P_s$ using orthographic projection. Then, we obtained the side view of $S_l$ (denoted by $C_l$), which is a point set located on the 2D plane. The concave hull algorithm proposed in [42] can be used to calculate the contour of an input point cloud, and it outputs an ordered set of points located at the boundary. We utilized the concave hull algorithm to extract the contour of $C_l$ with the maximum length among the resultant hull segment set as $d_s$. $P_l$ represents the contour of $C_l$ after the calculation of the concave hull. Connecting the points of $P_l$ in order, we constructed a 2D shape for use as the input of the 2D skeleton extraction algorithm. A sample contour $P_l$ and a constructed 2D shape are depicted in Figure 4a,b.

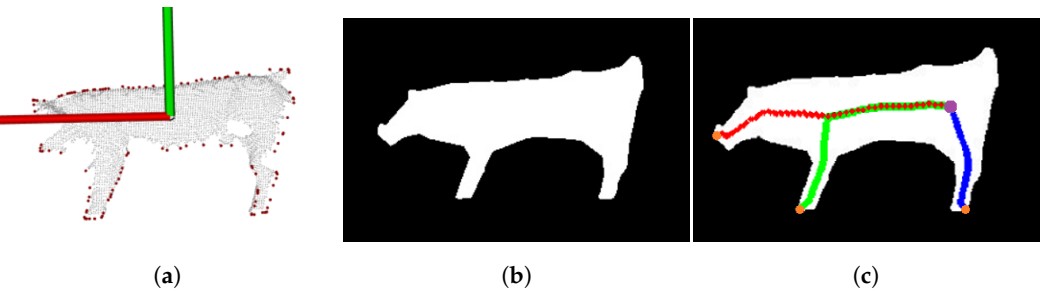

(**a**)　　　　　　　　　　　(**b**)　　　　　　　　　　　(**c**)

**Figure 4.** The visualization of a contour and a constructed 2D shape are depicted in (**a**,**b**). In figure (**a**), the contour points calculated by the concave hull algorithm are labelled in red. The skeleton extracted from the contour of a side view is shown in (**c**). The three branches are shown in red, green, and blue. The centre of the skeleton is labelled in purple, and the endpoints of the skeleton are labelled in orange.

### 2.3.2. Skeleton Extraction and Division

After obtaining the 2D shape constructed by $P_l$, we used the discrete skeleton evolution (DSE) algorithm [24] to extract the skeleton in 2D shapes. DSE is a method that performs skeleton pruning by iteratively removing the branches with the smallest relevance with regard to shape reconstruction. The stopping threshold and the number of branches are two parameters of DSE that can preserve required topology structure of the resulting skeleton. We set the stopping threshold and the number of branches of DSE as $d_t$ and $n_b$, respectively. Then, an accurate skeleton containing the main branches of the livestock was extracted from the 2D shapes using DSE. The centre point of the skeleton is the point with the maximum distance from the contour. The output skeleton can be divided into three branches from the centre point to the endpoints of the skeleton. An extracted skeleton composed of three branches is depicted in Figure 4c.

After obtaining the branches of the skeleton, we downsampled all the branches using an octree with leaf size $d_f$ to reduce the density of the skeleton points. Let $B_i, i \in \{1, 2, 3\}$ denote the downsampled branches. Since the extracted branches have no semantic information, we needed to identify the foreleg, hind leg, and torso from these three branches. Here we propose two methods to distinguish between these branches.

(i) Skeleton division based on detection (implement on pigs)

In the first method, we distinguished the branches by identifying the body parts based on deep learning. We used YOLOv4 [43] to detect the body part represented by the side view. YOLOv4 is a remarkable deep learning model for object detection. Combining the RGB information of the point set $C_l$ on the symmetry plane, we applied a perspective projection to create virtual RGB images. Then, we labelled these virtual images to build a training dataset. Figure 5 shows a virtual RGB image and the labelled areas of three body parts.

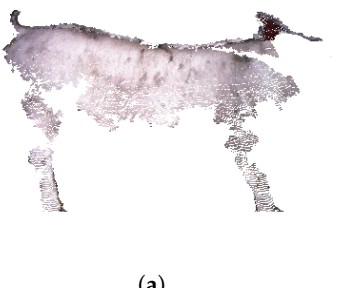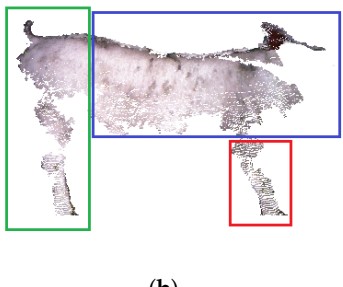

(**a**)                                        (**b**)

**Figure 5.** Virtual image (**a**) and labelled areas of three body parts (**b**). The areas of the body, foreleg, and hind leg are labelled in blue, red, and green, respectively.

We used the trained detector to determine the positions of the three body parts. The position of a body part is represented by a bounding box. The network predicted a probability score for each bounding box that indicated how likely the bounding box was to be located at the correct location. Multiple predicted bounding boxes may be generated for one body part. For each body part, we reserved the bounding box with the maximum score. As each branch was located at the centre of its corresponding body part, we utilized the bounding boxes to choose the branch. For an output bounding box, assuming that the number of points of $B_i$ in the bounding box is $N_i$, the number of all points of $B_i$ is $S_i, i \in \{1, 2, 3\}$. Then, we can calculate a ratio to determine which branch corresponds to this body part.

$$R_i = \frac{N_i}{S_i} \tag{1}$$

We chose the branch with the maximum $R_i$ for each body part. In addition, we can directly label a branch as missing when the bounding box of its corresponding body part is not detected.

(ii) Skeleton division based on spatial relationships

When the livestock data have no RGB information or its points are sparse, we can distinguish branches through the spatial relationships between branches. In the second method, we assumed that the torso branch point was not lower than the lowest point of the leg branches. Let $b_i$ denote the lowest point of $B_i$.

$$b_i = \arg_{b_j} \min\{b_j.y | b_j \in B_i\} \tag{2}$$

where $b_j.y$ is the y-coordinate value of $b_j$. The point $b_i$ with the largest y-coordinate value belongs to the torso branch. Then, we divided the remaining two branches into a foreleg branch and a hind leg branch according to the x-coordinates. $\bar{x}_i, i \in \{1, 2\}$ represent the mean values of the x-coordinates of all points in the leg branch.

$$\bar{x}_i = \frac{1}{N}\sum_{j=1}^{N} b_j.x \tag{3}$$

where $N$ is the number of points in the leg branch. The branch with the larger $\bar{x}$ corresponds to the foreleg branch and the other corresponds to the hind leg branch. Additionally, to address errors caused by missing data, we judged the validation of the leg skeleton according to the position of the centroid of the livestock. For instance, the average position of a hind leg skeleton point cannot be in front of the centroid. Similarly, the average position of a foreleg branch cannot be behind the centroid. In addition, if the angle between the leg branch and the ground is less than 30 degrees, we infer that this leg branch is invalid.

After distinguishing the branches, the leg branches usually include some points of the torso branch. Therefore, a leg branch extracted from a 2D shape may form a fold line,

such as the foreleg branch shown in Figure 4c. The reason for this is that the centre of the livestock skeleton, which is also the starting point of each branch, is usually located at the centre of the torso branch. Here, we used the RANSAC algorithm to extract lines from the leg branch that contained torso points. The sample model of the RANSAC algorithm was set to line with the distance threshold $d_l$. As the torso branch of a standing livestock is nearly parallel to the ground, we chose the line with the largest angle with the ground as the leg branch. Additionally, for the missing leg branch, we defined a leg branch at a proper initial position according to the leg skeleton points on the other side of the body. Adding a model for the leg of livestock with missing data can increase the applicability of the method in Section 2.4.1.

### 2.3.3. Calculation of the Leg Skeleton Position

Since the body surface is scanned from outside of the livestock, a large portion of the internal body data is missing. This results in the leg skeleton point not being moved to the local centre by ellipse fitting. Our method is to find the nearest banded point set for each leg skeleton point in its corresponding leg part. Then, we put each leg skeleton point in the outermost portion of its nearest banded point set and translate the leg skeleton point towards the inside of the body by the approximate distance of the leg radius $d^*$.

Specifically, we first translated all the leg skeleton points from the symmetry plane to the border of the livestock body along the $z$-axis. For each leg branch, we calculated its average x-coordinate value $\bar{x}$ and filtered the livestock within the range of $(\bar{x} - d_f, \bar{x} + d_f)$ to obtain a point set around this leg. If the leg branch was on the right side of the body, we identified the border point as the rightmost point of the point set, while the border point was the leftmost point if the leg was on the left side. Then, we translated each leg branch along the $z$-axis to the z-coordinate value of its corresponding border point, as shown in Figure 6a.

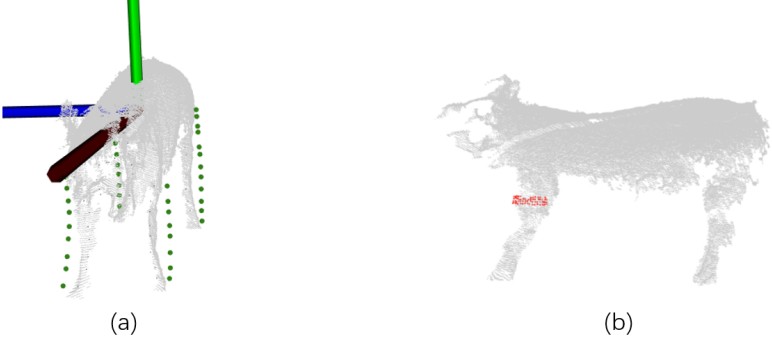

| (a) | (b) |

**Figure 6.** The leg branches moved to the border of the body (**a**) and the banded shape point set (**b**). The skeleton points are labelled in green and the banded shape point set is labelled in red.

Second, we calculated the banded point sets for all leg skeleton points by using the supervoxel clustering algorithm proposed in [44]. This algorithm divides a whole point cloud into regularly shaped clusters with specific sizes. We adjusted the seed resolution to $d_{sr}$ resulting in a proper scale for the supervoxel cluster. Let $C_i, i \in \{0, 1, 2 \cdots \}$ denote the resulting clusters. We defined the nearest cluster for each leg point according to the weighted squared distance between the leg point and the centroid of the cluster. The purpose of weighting is to prevent the closest cluster from being located in the head or other positions due to body twisting. For a leg point $p_i$, its weighted squared distance from a cluster $C_j$ can be calculated as:

$$d(p_i, C_j) = a * (p_i.x - c_j.x)^2 + b * (p_i.y - c_j.y)^2 + c * (p_i.z - c_j.z)^2 \tag{4}$$

where $c_j$ is the centroid of the cluster $C_j$. Then, for the leg skeleton point $p_i$, the closest cluster $C_k$ can be calculated as:

$$C_k = \arg_{C_i} \min d(p_i, C_i), i \in \{0, 1, 2 \cdots\} \tag{5}$$

We assumed that the adjacent clusters of $C_k$ are $\{C_0, C_1, C_2 \cdots, C_n\}$. To make the corresponding point set cover the width of the leg, we added all the adjacent clusters to the supervoxel clusters that had corresponding leg points. For the leg point $p_i$, its corresponding point set can be denoted as $\{C_k, C_0, C_1, C_2 \cdots, C_n\}$. The banded point set of $p_i$ can be calculated by pass-through filtering of its corresponding point set within the range of $(p_i.y - 3r, p_i.y + 3r)$. Figure 6b shows the visualization of a banded shape point set on the leg of the livestock.

Finally, we translated each leg skeleton point along the *z*-axis to the position that was equal to the z-coordinate value of the outermost point of its nearest banded point set. For the banded shape point set on the left side of the body, its outermost point is the point with the smallest z-coordinate value. The outermost point is the point with the largest z-coordinate for the banded shape point set on the right side of the body. Then, we moved the leg skeleton point towards the inside of the body by the distance $d^*$ to reach the final position.

### 2.3.4. Calculation of the Torso Skeleton Position

We combined the torso skeleton points extracted from the 2D contour via symmetry and the top view to calculate the final 3D torso skeleton points. To obtain the torso skeleton of the top view, the same 2D shape construction procedure for obtaining the contour of the side view in Section 2.3.1 was applied. $S_u = \{p_i | p_i.y > 0, p_i \in S\}$ represents the point set above the centroid of the livestock. We projected each point of $S_u$ onto a plane parallel to the ground and calculated the contour of the top view by the concave hull algorithm. We extracted the 2D skeleton of the top view contour by DSE and then down sampled the resulting skeleton with leaf size $d_f$ to obtain the torso branch denoted by $B_t$. The point set of the concave hull and the 2D shape containing the torso skeleton are shown in Figure 7.

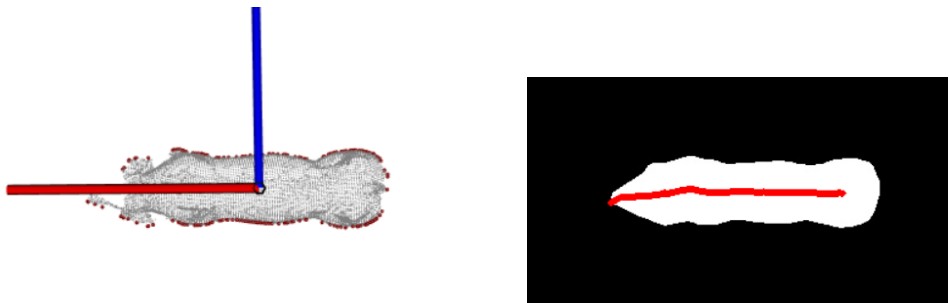

**Figure 7.** The constructed concave hull and the extracted torso skeleton for the top view of the livestock.

The torso branches of the side views were distinguished from the 2D skeleton on the symmetry plane in Section 2.3.2. Let $B_l$ and $B_r$ denote the torso branch of the left side view and right side view, respectively. For each torso point $b_l$ of $B_l$, we found a torso point $b_r$ from $B_r$ that satisfied:

$$\arg_{b_r} \min |b_l.x - b_r.x|, b_r \in B_r \tag{6}$$

Then, we used the midpoint of $b_l$ and $b_r$ as the integrated torso point. Let $B_s$ denote the integrated torso branch. Then, we used the same method to integrate the torso branch into $B_t$ and $B_s$. For each point $p_i$ in $B_s$, we calculated the point $p_j$ with the closest x-coordinate value to that of $p_i$ in the point set $B_t$.

$$\arg_{p_j} \min |p_j.x - p_i.x|, p_j \in B_t \tag{7}$$

Then, we obtained the torso branch in 3D space by adding the z-coordinate of $p_j$ to $p_i$.

After all the skeleton points were calculated, we sorted the points of the torso branch from head to tail in descending order of their x-coordinate values and connected all the points. Similarly, we sorted the skeleton points of each leg branch in descending order of their y-coordinate values and connected them. The highest point of the leg branch was close to the torso branch in height. For the connection between the leg branch and torso branch, we identified the two torso skeleton points that had the smallest distances from the highest points of the legs on both sides to connect the forelegs and hind legs, respectively. $B_f$ denotes the torso point between two forelegs, $L_l$ and $L_r$ denote the highest points of the left foreleg and right foreleg, respectively, and the torso point $B_f$ is calculated as:

$$B_f = \arg_{p_i} \min |d_e(p_i, L_l) + d_e(p_i, L_r)|, p_i \in B_s \tag{8}$$

$d_e$ denotes the Euclidean distance between two points. Then, we connected $B_f$ to $L_l$ and $L_r$. The connection process between the hind legs and the torso was the same.

### 2.4. Experimental Data and Posture Evaluation Application

The result of our curve skeleton extraction method can be directly applied to pose-related applications. An appealing feature of our extracted curve skeleton is the fact that each skeleton point belongs to a specific branch, and we can easily obtain the positions of the specified leg skeleton points. To demonstrate the potential of our method, we show how the extracted skeleton can be used to evaluate the posture of livestock [6] in the following section.

#### 2.4.1. Evaluation of Correct Body Posture Measurement

The 3D scanning data of a live pig are dynamic point cloud sequences containing its various postures. Most of the features can be measured accurately only if the pig has the correct posture. Therefore, choosing the frames with the correct postures from sequences is essential for a body measurement system. According to the assumptions made in [6,45], the requirements for a correct posture can be briefly summarized as: the four hooves of the measured pig must make a rectangle and the torso branch must be almost a straight line. In [6] the positions of four hooves were obtained by filtering the point set near the ground plane. However, this approach fails to acquire the correct quadrilateral formed by hooves when handling scanned data with missing legs. Our extracted skeleton provides a better way to evaluate the posture obtained from body measurements. We can directly find the four hooves from the lowest position of each leg branch to form a quadrilateral. The missing leg branch is labelled, and a model leg is added later in the appropriate position. The position where the model leg is added is calculated according to the leg on the other side of the body or a torso skeleton point, as shown in Figure 8.

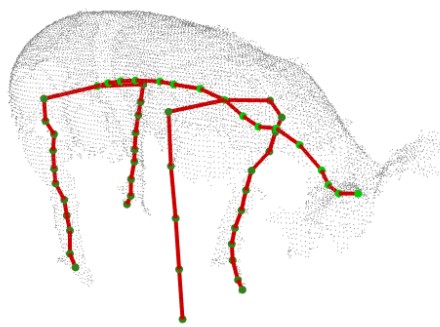

**Figure 8.** An extracted skeleton with an added model leg.

Although the model-derived added leg may not be in the position corresponding to the original part, it still provides an approximate quadrilateral for evaluating the posture of livestock.

Let us start with the extracted leg skeleton branches of a pig denoted by $L_i$, $i \in \{1, 2, 3, 4\}$. $L_1$, $L_2$, $L_3$, and $L_4$ correspond to the skeleton branches of the left foreleg, right foreleg, right hind leg, and left hind leg, respectively. First, we obtain the hoof point $h_i$ from $L_i$ by retrieving the point at the end of the branch. Second, we obtain the proper rectangle by connecting the points of the four hooves in clockwise and construct the vector form of each edge of the rectangle denoted by $v_i$. $v_i = h_{i+1} - h_i, i \in \{1, 2, 3\}$, and $v_4 = h_1 - h_4$, where $h_i$ is the vector form of the hoof point. Finally, we calculate the angle between adjacent vectors. Let $\alpha_i$ denote the angle located at hoof point $h_i$. The angles located at the vertices of the rectangle can be calculated by Equation (9).

$$
\alpha_i = \frac{v_i \cdot v_{i+1}}{|v_i||v_{i+1}|}, i \in \{1, 2, 3\},
$$

$$
\alpha_4 = \frac{v_4 \cdot v_1}{|v_4||v_1|},
$$

$$
S_1 = \sum_{i=1}^{4} |\alpha_i - \pi/2|
$$

(9)

The evaluation of the rectangle is given by $S_1$. The posture of the measured livestock is correct when the value of $S_1$ is less than a certain threshold $T_1$. Notably, $\alpha_i$ is not the interior angle of the rectangle, but the calculated $S_1$ is the same as that of the interior angle.

For the latter condition in which the torso branch is almost a straight line, we can define a measurement similar to $S_1$. We uniformly retrieve points from the sorted point set. We assume that the size of the measured torso skeleton is $N$ and the number of points to be retrieved is $k$. Then the retrieved index is $\lfloor i * \frac{N}{k} \rfloor, i \in \{0, 1, \cdots, k-1\}$. Let $B^* = \{p_j\}, j \in \{1, 2, \cdots k\}$ denote the retrieved point set. We present the following formula to estimate the straightness of the torso skeleton:

$$
\bar{y}_t = \frac{1}{k} \sum_{i=1}^{k} p_j.y, p_j \in B^*
$$

$$
S_2 = \frac{1}{k} \sum_{i=1}^{k} |p_j.y - \bar{y}_t|, p_j \in B^*
$$

(10)

where $p_j.y$ is the value of the y-coordinate of $p_j$. $y_t$ is the mean y-coordinate value of the points in $B^*$. When the value of $S_2$ exceeds a certain threshold $T_2$, this indicates that the body of the livestock is bent too much.

## 3. Results

We demonstrate the effectiveness of our method on point clouds containing multiple species of animals. The values of the parameters mentioned in Section 2.3 are listed in Table 1. Specifically, the size limitation $d_s$ determines the accuracy of the contour. Smaller values can be used to capture more details of the contour. However, when the value of $d_s$ is too small, holes form inside the extracted contour. Multiplying by $x_r$ enables $d_s$ to change dynamically according to the scale of the livestock. In the DSE algorithm, the stopping threshold $d_t$ relates to the importance of the branch in the 2D shape. When $d_t$ decreases, shorter branches are retained. The number of branches $n_b$ limits the number of output skeleton branches. According to our test, the values of $d_t$ and $n_b$ listed in Table 1 can retain the required number of branches for most of the scanned pigs. The projected profiles of different species are different. For the dataset of the other species, we apply different values of $d_t$, $n_b$, and $d_s$ to obtain skeleton branches with the desired topological structure. The optimal parameters for different species are listed in Table 2. The octree is used to down sample the skeleton points, and its leaf size $d_f$ defines the density of the skeleton points.

The larger the leaf size $d_f$ is, the sparser the skeleton points. The RANSAC algorithm with the distance threshold $d_l$ can extract lines from the fold line that consists of a leg branch and some torso points. The seed resolution $d_{sr}$ controls the supervoxel cluster with a proper scale. The weight values a, b, and c of distance $d(p_i, C_j)$ guarantee that the closest cluster of each leg skeleton is located on the leg point cloud. $d^*$ is the distance of the leg branches moving inward, which is an estimation of the radius of the pig's leg. These parameters are held constant in the experiment when dealing with scanned datasets. We used the default parameters of the $L_1$-median [28] method during the comparison. Since Point2skeleton [32] is an unsupervised learning method, this paper divides 200 pig body point clouds into a training set and a test set in the ratio of 8 to 2. After training on the training set of 160 point clouds, the results are tested on the test set of 40 point clouds.

**Table 1.** The parameters discussed in Section 2.3. $r$ is the resolution of the input point cloud. $x_r$ is the range of the x-coordinate value of the livestock data. $d_s$ is the maximum segment of the concave hull. $d_t$ and $n_b$ denote the stopping threshold and the number of branches of DSE, respectively. $d_f$ is the leaf size of the octree. $d_l$ is the distance threshold of the RANSAC algorithm for extracting the leg branch. $d_{sr}$ is the seed resolution of the supervoxel algorithm. a, b, and c are weight values of distance $d(p_i, C_j)$. $d^*$ is the distance of the leg branches moving inward.

| Parameter | $d_s$ | $d_t$ | $n_b$ | $d_f$ | $d_l$ | $d_{sr}$ | a | b | c | $d^*$ |
|---|---|---|---|---|---|---|---|---|---|---|
| Value | $16r * x_r$ | 0.003 | 5 | $6r$ | $10r$ | 0.5 | 1 | 1.5 | 0.5 | $4r$ |

**Table 2.** Different values of parameters used in animal models.

| Parameters | Hippo | Water Buffalo | Cow | Rhino | Horse | Cattle |
|---|---|---|---|---|---|---|
| $n_b$ | 5 | 6 | 5 | 5 | 5 | 5 |
| $d_t$ | 0.003 | 0.001 | 0.001 | 0.08 | 0.003 | 0.003 |
| $d_s$ | 0.07 | 0.07 | 0.08 | 0.08 | 0.2 | 0.06 |

Notably, due to the limited amount of acquired data, part of the skeleton distinguished by the detector is from the training set. Specifically, 140 point clouds from 14 pigs were randomly selected as the training set. The remaining 60 point clouds were used as the test set. For each point cloud of the training set, we created two virtual images from two sides of the livestock. A total of 280 virtual images were created. We then applied the vertical flip and horizontal flip to augment the dataset to 840 virtual images. We proved the validity of the detection model on the test set, and the results show that all body parts of the livestock data in the test set were correctly detected. The detector can provide a stable body position for our processing pipeline. Therefore, we infer that it has little influence on the results of the whole algorithm.

*3.1. Curve Skeleton Extraction*

The correctness of our algorithm relies on the correctness of the calculated concave hull and the topology of the skeleton calculated by the DSE algorithm. However, as the data are affected by the data quality and the poses of livestock, there is no strict guarantee that the default parameter settings can calculate all the data correctly. Due to the complexity of the errors, it is difficult to evaluate the robustness of the algorithm. To evaluate the algorithm quantitatively, we summarize and define the errors as the detection error and connection error. Since the properties of these two methods are different, the causes of these errors are different. Therefore, the definitions of these two kinds of errors are based on the similarity and potential consequences of errors. Specifically, the complete skeleton consists of a torso branch and four leg branches that are extracted from the main body parts of livestock. Detection error means that the method fails to extract all the skeleton branches of the main body parts or the number of extracted branch points is too small. This results in the skeleton points being distributed in a small area of its corresponding body part, as

shown in Figure 9a,d,g. We regard an extracted skeleton branch with a number of points less than 5 as a detection error when the data of its corresponding body part are not missing. For our method, when a leg branch is labelled invalid because the branch extracted from the symmetry plane is wrong or the skeleton division is wrong, the incorrect branch is erased and replaced by a supplemental leg. Therefore, a leg that is supplementary but not detected when the leg data are not missing still counts as a detection error, as shown in Figure 9g. Connection error means that the extracted main branches fail to connect correctly, or the connections of a branch are bad. The correct connections between the branches should be consistent with the topology of livestock. Therefore, each leg branch should connect to the torso branch, with no connections between different leg branches.

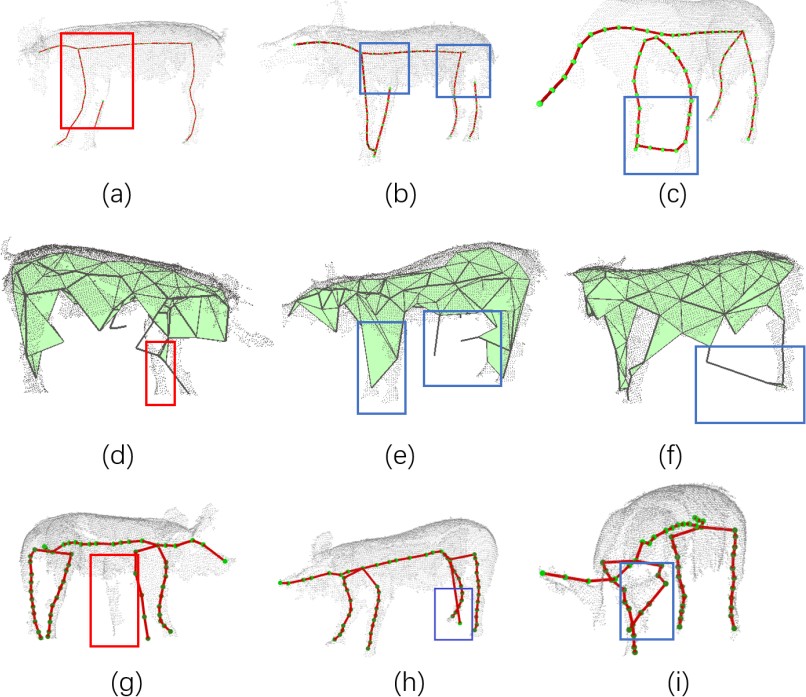

**Figure 9.** Detection errors and connection errors that occurred in the three methods. The results extracted by $L_1$-median method, Point2skeleton method and our method are listed in the first row (**a**–**c**), second row (**d**–**f**), and third row (**g**–**i**), respectively. Detection errors are labeled in red. Connection errors are labeled in blue.

Figure 9b,c,f show the connection error caused by failures when connecting the main branches in a correct topological structure. As each skeleton point extracted by our method belongs to a specific branch and all the skeleton points on the same branch are connected in order, a connection error is represented as a branch containing points that are severely out of position. As shown in Figure 9e,h,i, the incorrect points may be the outliers of a branch that breaks the smoothness of the total branches or the points in an incorrectly detected branch.

We evaluate the robustness of the algorithm by calculating the percentage of erroneous skeletons among all test results and the average number of errors per erroneous skeleton (ANE). The ANE is equal to the total number of errors divided by the number of skeletons with errors. We present quantitative results in Table 3. Projection-based curve skeleton (*PBCS*) stands for our method. In Section 2.3.2, we proposed two methods for skeleton division. *PBCS–S* is based on the spatial relationship, and *PBCS–D* is based on detection.

**Table 3.** Results of the comparison between the $L_1$-median skeleton approach and our curve skeleton extraction method. The percentage represents the percentage of data with errors out of the total data. ANE stands for the average number of errors per erroneous data point.

| Method | Detection Error | | Connection Error | |
|---|---|---|---|---|
| | Percentage (%) | ANE | Percentage (%) | ANE |
| *Point2skeleton*(40) | 0 | 0 | 52.5 | 2.33 |
| $L_1$-median | 20 | 1.3 | 74 | 1.61 |
| *PBCS–S* | 8.5 | 1.24 | 11 | 1.14 |
| *PBCS–D* | 10.5 | 1.1 | 2.5 | 1.2 |

The detection error rate of our method is slightly lower than that of the $L_1$-median approach, and the average number of errors per erroneous skeleton is smaller than that of the $L_1$-median approach. On average, 1.3 branches cannot be detected in the erroneous data extracted by the $L_1$-median method, while the ANE of *PBCS–S* equals 1.24 and the ANE of *PBCS–D* equals 1.1. The detection error rate of *Point2skeleton* is 0, which means it can calculate all skeleton points from a given livestock point cloud. However, the connection error rate of the *Point2skeleton* and $L_1$-median method are much larger than that of our method. The percentage of the skeletons with connection errors extracted by *Point2skeleton* approach reaches 52.5%, and there is an average of 2.33 errors per erroneous result. The connection error rate and the ANE calculated by the $L_1$-median are 74% and 1.61, respectively. The connection error rate and the ANE calculated by *PBCS–S* are 13.5% and 1.11, respectively. For *PBCS–D*, the percentage of connection errors and the ANE are 2.5% and 1.2, respectively. This shows that our method significantly outperforms the $L_1$-median and *Point2skeleton* in preserving the correct topological structure of pig point clouds.

This performance comes from our top-down design. Specifically, the skeleton extracted from the contour preserves the correct topology well, and the separate processing for each side of the livestock enables our method to avoid connection errors between the legs. Compared with *PBCS–S*, *PBCS–D* further avoids connection errors. *PBCS–S* distinguishes different branches through their relative positions, while *PBCS–D* predicts a location range for each branch. Therefore, *PBCS–D* prevents the wrong branch from being considered a branch of the main body part. In our method, errors mainly occur during the construction of the contour and the extraction of the 2D skeleton. The contour of the side view is captured by the concave hull algorithm. When the concave hull forms holes in the contour, making the contour severely irregular, the position of the 2D skeleton extracted by DSE severely deviates or even becomes amiss. Additionally, DSE cannot ensure that all extracted 2D skeleton branches correspond to the main body parts of the pig. Stabilizing the contour construction and the 2D skeleton extraction processes can improve the robustness of our method.

Even if our curve skeleton satisfies the thinness requirement, its smoothness is not strictly guaranteed. Figure 10 shows an example of a skeleton of a live pig effectively extracted by our method and another extracted by the $L_1$-median method to visually demonstrate the aforementioned difference in smoothness. Intuitively, the skeleton extracted by the $L_1$-median method is smoother, especially the connection between the leg branch and torso branch. Additionally, $L_1$-median utilizes ellipse fitting to improve the centrality of the skeleton. However, such a recentering step is not the right solution for raw scanned point clouds characterized by missing areas. The ellipse fitting algorithm requests the local cylindrical shape, and this request cannot always be satisfied by raw scanned pig point clouds due to missing data. In Section 2.3.3, we constructed the banded shape for each leg point by pass-through filtering the nearest supervoxel and its adjacency, and this also provides an excellent way to find a proper subset of point clouds for performing recentering.

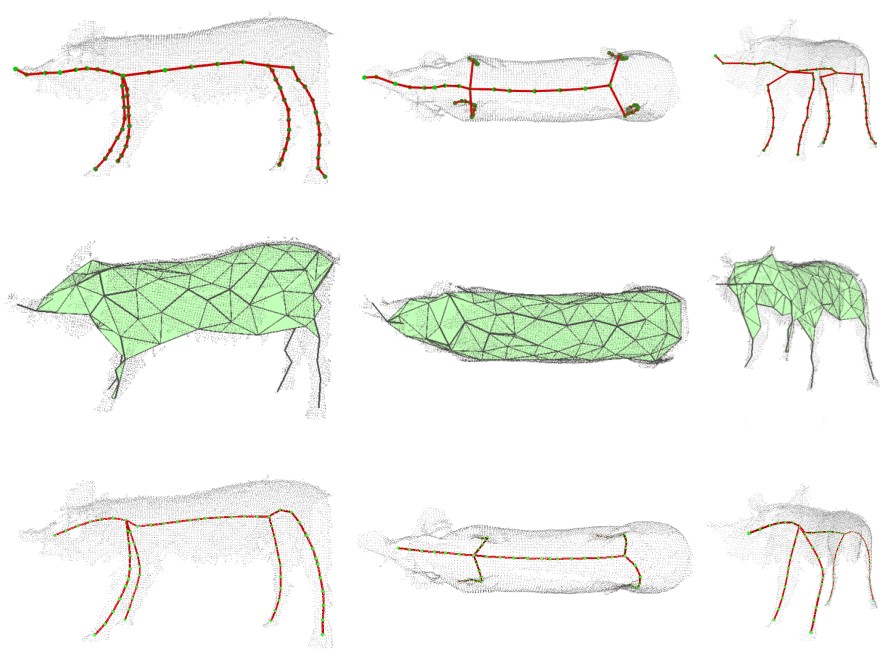

**Figure 10.** Visualization of a skeleton of a live pig effectively extracted by our method and another extracted by the *L*1-median method. The pictures in the first line show the skeleton extracted by our method, and the pictures in the second line show the skeleton extracted by *L*1-median.

### 3.2. Results of Posture Evaluation

In this section, based on our extracted skeleton, we evaluate the postures of the 20 Landrace pig sequences mentioned above. Since our method is susceptible to incorrect scanning postures, we formulate a selection scheme to ensure that the correct measurement posture is selected. We use $S_1$ in Section 2.4.1 to evaluate the postures and set the threshold $T_1$ to 1.5. In a sequence, we first consider the frames that are not labelled missing a leg. If there are frames with $S_1$ values that are lower than $T_1$, we choose the frame with the smallest $S_1$ among these frames. When there is no unlabelled frame satisfying the threshold, we use the same criteria to select from the frames with one missing leg. If there is no satisfying frame in a sequence, no frame is selected. Out of the 20 total sequences, we select data with correct postures for 17 sequences in this manner. Several pigs with postures leading to different $S_1$ values are shown in Figure 11.

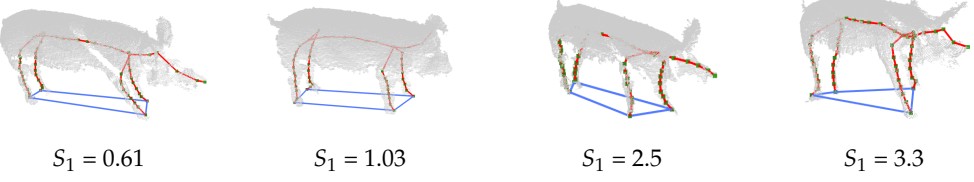

$S_1 = 0.61$        $S_1 = 1.03$        $S_1 = 2.5$        $S_1 = 3.3$

**Figure 11.** Illustration of pigs with different postures and their corresponding S1 values.

### 3.3. Results and Comparison with Other Animals' References

From the above experimental results, we can see that the detection accuracy of our method is 91.5% when using the method based on spatial relationship and 89.5% when using the method based on computer detection, which is significantly higher than that of L1-median method, but lower than *Point*2*skeleton* method. However, the connection error of our method is 11.5% using spatial relationship and 2.5% using detection, while the connection error of other methods are over 50%. Furthermore, We test our method on the synthetic dataset to verify its applicability for different species. As shown in Figure 12,

compared to *Point2skeleton* and $L_1$-median, our method can preserve the topological structures of different species. For the horse point cloud, the constructed 2D shape cannot outline the contour between the hind leg and the tail, resulting in the leg branch of the extracted skeleton being out of position. Landmark detection provides a way to estimate the position of the tail, and segmenting the tail before processing may be a good way to handle this situation.

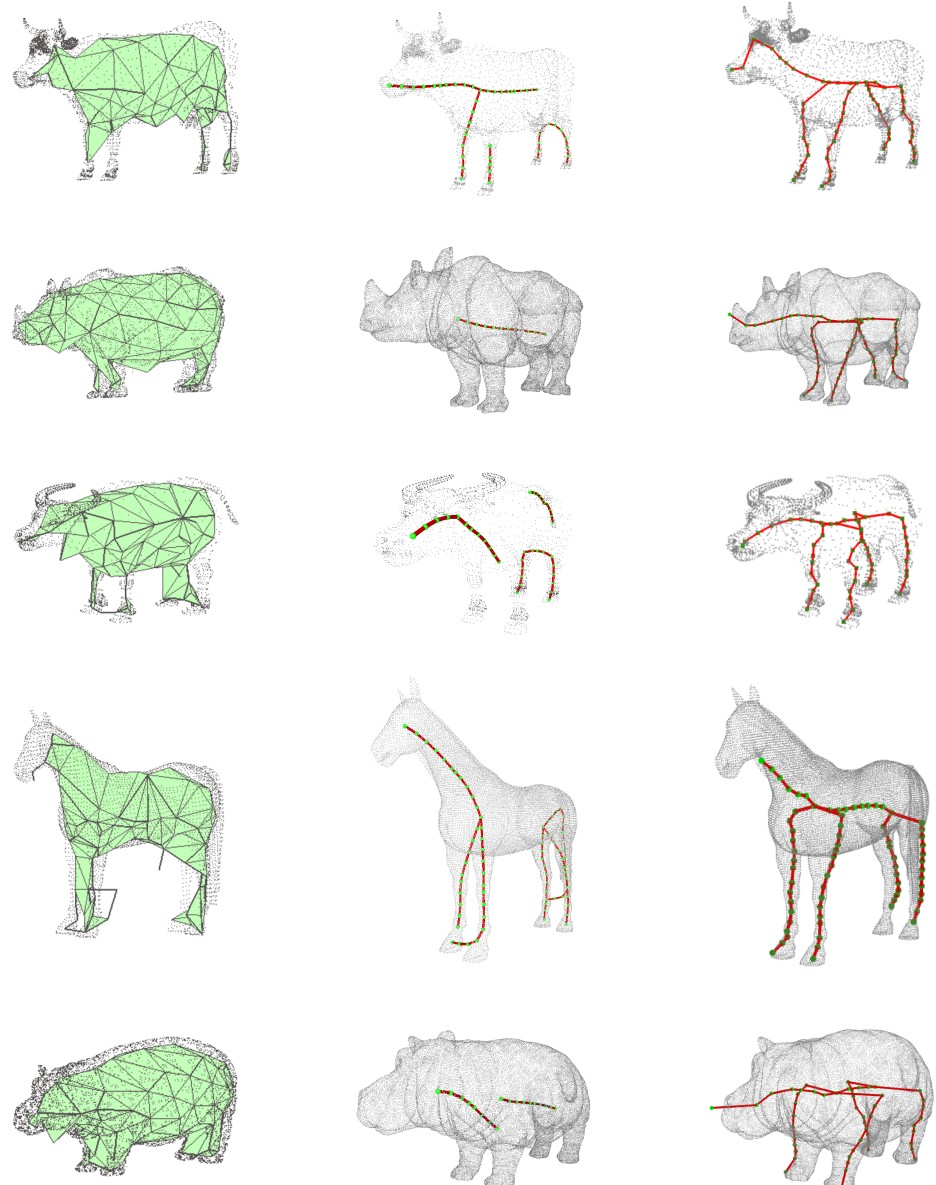

**Figure 12.** Curve skeletons of different species extracted by the three methods. The results of *Point2skeleton*, $L_1$-median, and our method are displayed in the first, second, and third columns, respectively.

## 4. Conclusions

In this paper, a novel curve skeleton extraction method specifically designed for incomplete point clouds of livestock has been proposed. Compared to the well-known and general skeleton extraction methods, connection errors are mostly avoided in our results; furthermore, our algorithm is efficient and has important advantages in terms of processing livestock point clouds with a larger number of points. The extracted skeleton is

able to evaluate the posture of the raw scanned pig, and this can assist in selecting data suitable for body size measurements. The algorithm was also tested for other quadrupeds, and we extracted skeletons with the correct topologies. However, the skeletons extracted from live cattle data have a high error rate, which is mainly caused by the process of 2D skeleton extraction and skeleton division. A shortcoming associated with our processing pipeline is its reliance on parameters. When dealing with animals of different species, some parameters in our algorithm need to be adjusted.

We tried using a detection-based deep learning method for skeleton division to improve the robustness of our method. This improvement reduces the error rate on the pig dataset. However, more data of other species need to be acquired and we need to improve the data acquisition system. As an efficient way of reducing dependencies and optimizing the processing pipeline, improvements based on deep learning methods will be investigated in the future. To make our method more applicable to motion-related tasks, we need to further improve the robustness and accuracy of our method. Future research will also include skeleton-based behavioural analyses to extract valuable information related to the health status of animals.

**Author Contributions:** Conceptualization, H.G.; methodology, Y.H. and A.D.; software, X.L. and Z.G.; validation, X.L. and Z.G.; formal analysis, Y.H. and H.G.; investigation, Y.H.; resources, H.G., A.P. and A.R.; data curation, A.R.; writing—original draft preparation, Y.H., A.D. and A.P.; writing—review and editing, F.M. and A.P.; visualization, A.D.; supervision, H.G.; project administration, H.G.; funding acquisition, H.G. and A.R. All authors have read and agreed to the published version of the manuscript.

**Funding:** This research was funded by the National Natural Science Foundation of China [grant numbers 42071449, 41601491] and the Russian Science Foundation, grant no. 21-76-20014.

**Institutional Review Board Statement:** Not applicable.

**Informed Consent Statement:** Not applicable.

**Data Availability Statement:** Data are available from the first author.

**Acknowledgments:** The authors wish to thank everyone who participated in the software assessment. We thank the Shang-Dong WeiHai swine-breeding centre of DA BEI NONG GROUP for providing us with materials for the experiment.

**Conflicts of Interest:** The authors declare no conflict of interest.

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
