# Peer review of "Curve Skeleton Extraction from Incomplete Point Clouds of Livestock and Its Application in Posture Evaluation"

_agriculture, doi:10.3390/agriculture12070998_

Round 1

Reviewer 1 Report

This is a very interesting and technical article. It is well written and easy to follow. The work is rigorous. Note that I am not a specialist of skeleton models and I’m not able to judge the novelty of the method and the quality of the bibliography, although it seems to be well justified. My only major comments is on the presentation of the dataset. Could you provide more details, do you have variability within your dataset? Variability in animal shape, background, and angle of view? Does the pictures are well spaced in time or just separated from few seconds? It is important to understand how the training and test set are different. I also not so sure of the interested of using a 3D camera, what king of extra information is provided by the point cloud?

Minor comments
L.110 Do you have any recommendation on the relative position of the animal from the camera? Angle of view? Does it work only from side view?
L.207 Specify how many images for training and testing, as well as the results of the evaluation.
l.430 We selcetED
Discussion. Need some words on computing time and possible application in the filed.

Reviewer 2 Report

The study is interesting, but some revisions should be made.

1) What is your contribution? This and the general problem should be written clearly. It is as if the methods and approaches in the literature were used in many places in the proposed method section. What is your original and new approach? Or did you combine existing methods?

2) How can you prove that the branches used were chosen from the right place?

3) How does the position of animals, such as standing, sitting, and sleeping, affect your results? What role does it play in your fairness?

4) Do shadows have a negative effect on your method? If yes, what is your solution?

5) Can you compare your approach with the metaheuristics approach in the extraction phase? especially in terms of cost and accuracy.

6) Is the experimental data you use open source? I didn't see any reference. How reliable is it?  Also, what features does this data have? 

7) Your method should be supported with numerical parameters in the results. In addition, it should be compared with some methods in the literature.

8) Grammar should be seriously reviewed.

9) Figures could be presented better especially in quality and resolution.

Round 2

Reviewer 2 Report

Full, clear and correct answers were given to the requested and written comments. Therefore, my opinion is that it should be accepted.